# Relation between motor competence and academic achievement: The mediating role of fitness and cognition in boys and girls

Antonio Fernández-Sánchez[1,2], Abel Ruiz-Hermosa[1,2,3]*, Andrés Redondo-Tébar[1,4], Ana Díez-Fernández[1,5], Vicente Martínez-Vizcaíno[1,6], María Eugenia Visier-Alfonso[1,5], Mairena Sánchez-López[1,2]

1 Social and Health Research Center, Universidad de Castilla-La Mancha, Cuenca, Spain, 2 Faculty of Education, Universidad de Castilla-La Mancha, Ciudad Real, Spain, 3 Faculty of Sport Sciences, ACAFYDE, Universidad de Extremadura, Cáceres, Spain, 4 Faculty of Nursing, University of Castilla-La Mancha, Ciudad Real, Spain, 5 Faculty of Nursing, University of Castilla-La Mancha, Cuenca, Spain, 6 Faculty of Health Sciences, Autonomous University of Chile, Talca, Chile

* abel.ruizhermosa@uclm.es

## Abstract

### Introduction

Gross motor competence is positively associated with academic achievement in schoolchildren, potentially mediated by fitness and cognition. However, the extent to which these mediators contribute—and whether effects differ by sex—remains unclear. This study explored the mediating roles of specific fitness components and executive function in the relationship between gross motor competence and academic achievement, considering sex differences.

### Methods

This cross-sectional study included 562 Spanish schoolchildren aged 9–11 years (293 girls). Gross motor competence was evaluated using the Movement Assessment Battery for Children-Second Edition; fitness components (cardiorespiratory fitness, speed/agility, upper and lower body strength) through the ALPHA-Fitness test battery; executive function using the NIH Toolbox Battery; and academic achievement from school grades in language and mathematics. Serial multiple mediation models were applied using the PROCESS macro in SPSS, adjusted for age, BMI, and maternal education level. Analyses were conducted for the total sample and by sex.

### Results

Both fitness and executive function partially mediated the relationship between gross motor competence and academic achievement. In the total sample, direct effects explained most of the association (51–73%), followed by the cognitive pathway (20–31%), fitness pathway (11–19%), and multiple pathway—gross motor competence,

**Data availability statement:** The minimal anonymized dataset required to replicate our study findings has been published in the ZENODO public repository and is openly accessible at the following URL: https://zenodo.org/records/17070143 (DOI: 10.5281/zenodo.17070143).

**Funding:** The Ministry of Economy and Competitiveness-Carlos III Health Institute (FIS PI16/01919) funded this study. Additional funding was obtained from the Research Network on Preventative Activities and Health Promotion (RD12/0005/0009). Furthermore, Andrés Redondo-Tébar and Abel Ruiz-Hermosa are postdoctoral researchers funded by the Margarita Salas Fellowship through the University of Castilla-La Mancha "Next Generation EU" (2022-POST-21124 and 2021-MS-20547, respectively). The funders had no role in study design, data collection and analysis, decision to publish, or preparation of the manuscript.

**Competing interests:** The authors have declared that no competing interests exist.

**Abbreviations:** MC, motor competence; GMC, gross motor competence; EF, executive function; CRF, cardiorespiratory fitness; S/A, speed/agility; UBS, upper body strength; LBS, lower body strength; AA, academic achievement; IE, indirect effect.

fitness, executive function, and academic achievement—(4–9%). Sex-specific analyses showed that cognitive mediation was predominant in boys, accounting for over half of the total effect (56–69%), with no direct effect observed. In contrast, fitness mediation was more relevant in girls, especially through cardiorespiratory fitness and speed/agility, each contributing up to 20% of the effect. The multiple pathway was also significant in girls.

## Conclusions

Enhancing motor competence may improve academic outcomes, partly through gains in fitness and executive function. These findings support implementing integrated school programs, tailored to sex-specific needs—emphasizing cognitively engaging activities for boys and fitness-focused strategies for girls. The cross-sectional design implies association, not causality.

## Introduction

Motor competence (MC) can be defined as a person's ability to execute different motor acts, including coordination of fine and gross motor skills that are necessary to manage everyday tasks [1]. There is a growing body of evidence suggesting a positive association between MC and physical health outcomes, including physical fitness and weight status [2]. In addition, MC has been widely recognized as a potential predictor of academic achievement (AA) [3,4]. However, the underlying mechanisms that drive this relationship have not been sufficiently elucidated.

Cognitive processes have been suggested as potential mediators between MC and AA [5]. This association may be attributed to shared neural pathways and overlapping cognitive demands between motor skills and executive function (EF) [6,7]. This is because the same brain regions are partially responsible for the development of MC and EF in childhood [8,9]. Despite this, few studies have examined the mediating role of EF in this relationship, suggesting that MC exerts a positive influence on AA indirectly through EF [5,10]. However, the evidence on sex differences remains inconclusive. Since one study reported no significant sex differences [11], another found that EF mediated the relationship exclusively in boys [12].

Given the well-established relationship between MC and physical fitness [13], as well as the documented positive association between fitness, cognitive function, and AA [3], it is reasonable to hypothesize that physical fitness could serve as a mediating variable in the relationship between MC and AA. However, to the best of our knowledge, only one study has directly examined this hypothesis, reporting no significant effect [14]. Moreover, although physical fitness is generally positively associated with AA [15], this relationship may vary depending on the specific fitness component considered. Cardiorespiratory fitness (CRF) and speed/agility (S/A) have consistently proven to have a positive effect on AA [16–18], but the role of muscular strength remains unclear [15,16]. Thus, it is reasonable to posit that the mediating effect of fitness on the MC-AA relationship may differ across fitness components.

Thus, the association between MC and AA may be explained through two principal pathways: the cognitive pathway and the fitness pathway. Furthermore, a third pathway may be proposed, termed the 'multiple pathway', which suggests that improvements in MC may result in enhancements in physical fitness, which could, in turn, enhance cognitive function and AA. This multiple pathway will be analyzed for the first time in this study, thereby providing a new and original viewpoint on the relationship between MC and AA.

Finally, research indicates that boys and girls exhibit distinct trajectories in physical and cognitive development [19], as well as in brain activation during cognitive tasks [20]. Such differences could affect the relationship between MC and its connection to fitness, cognitive functions, and AA [21]. However, no studies have specifically analyzed the mediating role of fitness and EF by sex. Despite increasing evidence on the links between MC, fitness, EF, and AA, previous studies have relied on partial assessments, single-component fitness measures, and inconsistent analytical approaches. Critically, no research has examined multiple fitness components and EF simultaneously or explored sex-specific mediation pathways. This study addresses these gaps by providing a comprehensive analysis of how gross motor competence (GMC) relates to AA in boys and girls, using validated tests in a representative sample of Spanish schoolchildren aged 9–11 years.

It is therefore necessary to conduct further in-depth research into the relationships between MC, fitness components, cognitive abilities, and AA, with a particular focus on potential sex differences. The objective of this study was to examine whether fitness and EF act as mediators in the relationship between GMC and AA, and to identify potential differences between boys and girls. Based on the reviewed literature, it is hypothesized that EF and specific fitness components partially mediate the relationship between GMC and AA, and that the magnitude of these mediations differs between boys and girls. Given the cross-sectional design, the study focuses on identifying associations rather than establishing causal relationships.

## Materials and methods

### Study design and participants

The present study is a cross-sectional analysis of the baseline data from a cluster-randomized controlled trial (NCT03236337) to assess the effectiveness of a physical activity intervention program (MOVI-daFit!) on reducing fat mass and cardiovascular risk and improving physical fitness, EF, and AA among schoolchildren. Ten schools from ten towns in the province of Cuenca, Spain, participated in the study. In all schools, all children belonging to the fourth and fifth grades of primary school (9–11 years old) were invited to participate. The study design, sampling procedures and methods have been fully described elsewhere [22].

The study protocol was approved by The Clinical Research Ethics Committee of the 'Virgen de la Luz' Hospital in Cuenca (REG: 2016/PI021). Before data collection, an informational session was conducted in each classroom to explain the study's objectives, the voluntary nature of participation, the anonymity of responses, and the right to withdraw at any time without consequences. These aspects were communicated in age-appropriate language to ensure the children's understanding. Following this explanation, each child was explicitly asked to give their verbal consent, in accordance with the procedure approved by the ethics committee. This consent was given in the presence of the classroom teacher, the responsible researchers, and the children's peers, who served as witnesses. In addition, written informed consent was obtained from the parents or legal guardians of all participants.

For the analysis, only students with complete data—those who completed all assessment tests—were included. Students with disabilities were not excluded solely on the basis of disability; they were eligible if they met the inclusion criteria described in the original trial. Children were excluded if they exhibited difficulties in learning Spanish; if teachers or parents reported serious physical or mental disorders that could hinder participation in program activities; or if pediatricians identified chronic conditions—such as heart disease, diabetes, or asthma—that could prevent participation.

## Instruments and study variables

The recruitment period for this study began on September 11, 2017, and ended on October 26, 2017. All measurements were conducted by trained researchers who followed standardized protocols to ensure consistency and accuracy in data collection. The evaluators were blinded to the participants' group assignments.

## Gross motor competence

Gross motor competence refers to the skilled performance of a wide range of motor tasks, as well as the coordination and control of movements involving large muscle groups [23] and is a prerequisite for child physical activity participation and also for engagement in learning and social activities, including sports and games [24]. To assess GMC, two dimensions (aiming-catching and balance skills) of the validated Spanish version of the Movement Assessment Battery for Children-Second Edition (MABC-2) were used [1] for the age ranges 2 (7–10 years) and 3 (11–16 years). This battery has shown strong content and construct validity for assessing MC in children, with test–retest reliability coefficients above 0.80 [1]. The tasks performed were: 1) Aiming-catching 1 *(catching)*: The child had to throw a tennis ball against the wall and catch it with both hands (9–10 years); The child had to throw a tennis ball against the wall and then catch it using only one hand (11 years). 2) Aiming-catching 2 *(throwing)*: The child had to aim a beanbag into a red circle on a mat (9–10 years); The child had to throw a tennis ball into a red circle on the wall (11 years). 3) Balance 1 *(static)*: Single-board balance – The child had to balance on one foot on the balance board (9–10 years); Two-board balance – The child had to balance on the balance board, ensuring that the heel of one foot and the toes of the other foot touch (11 years). 4) Balance 2 *(dynamic)*: Heel to toe – The child had to walk along a line while the heel of one foot touches the toes of the other foot. (9–10 years); Walking backwards heel to toe – The child had to walk backwards along a line, making sure that the toes of one foot touch the heel of the other (11 years). 5) Balance 3 *(dynamic)*: Hopscotch – The child had to jump forward on one leg from mat to mat starting from a standing position (9–10 years); Zigzag hopping – The child had to jump diagonally from one mat to another on one leg (11 years) [25]. Each test received a raw score (number of catches, number of hits on target, seconds on one leg, number of steps on the line and number of right jumps, respectively) where a higher score indicated a better GMC. To enable comparison between tests with different measurement units, raw scores for each test (aiming-catching, static balance, and dynamic balance) were standardized into z-scores. Z-scores were calculated within age- and sex-specific groups by subtracting the group mean from the individual score and dividing by the group standard deviation. The GMC Total Score was obtained as the sum of all standardized test scores.

## Fitness variables

Fitness is defined as a set of attributes related to a person's ability to perform physical activities that require aerobic capacity, endurance, strength, or flexibility and is determined mostly by a combination of regular activity and genetically inherited ability [26]. The evidence-based ALPHA-Fitness test battery [27], designed to assess physical fitness, was used. This battery has demonstrated high criterion validity and reliability across European school populations. Specifically, the 20-meter shuttle run, 4 × 10 meter shuttle run, handgrip strength test, and standing broad jump test have all been validated as reliable field-based assessments of CRF, S/A, and muscular strength, respectively [27]. The following tests were performed:

**Cardiorespiratory fitness,** using the 20-metre shuttle run test, a valid test to estimate aerobic capacity in children. The children were required to run 20 meters between two lines while keeping pace with the rhythm marked by acoustic signals. The initial speed of 8.5 km/h was increased by 0.5 km/h for each stage (one stage equals one minute). The last half stage completed by the child was considered an indicator of CRF. Maximal oxygen intake ($VO_2$max) was estimated by applying the Léger formula [28].

**Speed-agility,** using the 4 × 10 meter shuttle run test. Children were required to run and turn (shuttle) 10 meters twice at maximum speed (4x10m). Two parallel lines were drawn on the floor. When the start was given, the children ran as

fast as possible to the other line and returned to the starting line twice, crossing both lines with both feet. Two attempts were made, with an interval of five minutes between attempts, and the best score was recorded in seconds. A lower score indicated a better S/A.

**Upper Body strength** (UBS), through the handgrip strength test, determined in kilograms using a digital dynamometer with adjustable grip TKK 5401 Grip-DW (Takeya, Tokyo, Japan). The handgrip strength test has high-to-very high construct validity with total muscle strength in healthy children, among other parameters [29]. The test was performed twice with each hand and the arithmetic mean of the four measurements was computed. Handgrip data was normalized by dividing absolute handgrip strength (kg) by body weight (kg).

**Lower Body strength** (LBS), using the standing broad jump test, which measures explosive LBS. The children jumped horizontally to achieve the maximum distance, and the best of three attempts was recorded in centimeters.

### Executive function

Executive function refers to a family of mental processes that have been categorized in three core functions: inhibitory control (the ability to selectively attend, focusing on what we choose and suppressing attention to other stimuli), cognitive flexibility (the ability to change perspectives) and working memory (the ability to hold information in mind and mentally working with it) [30]. Three EFs, using the NIH Toolbox software (NIH Toolbox in Spanish, v. 1.8) [31] were assessed. This tool has demonstrated high internal consistency (Cronbach's alpha > 0.80), excellent test–retest reliability, and strong convergent validity with standardized neuropsychological measures in pediatric populations [31,32]. All tests were performed individually in a quiet classroom using a tablet (iPad Pro, Apple, Inc., California, USA). The evaluation tests included in the NIH Toolbox for each cognitive domain are listed as follows:

**Inhibitory control,** using an adaptation of the Flanker test [32]. Participants had to indicate the left-right direction of a centrally shown stimulus while inhibiting their attention to the potentially incongruent stimuli around it. In some trials, the orientation of the flanking stimuli was congruent with the orientation of the central stimulus (>>>>> or <<<<<,) while in other trials, this orientation was incongruent (>> <>> or <<> <<). The NIH Toolbox contained a practice block of trials. If participants passed it, a 20-trial block was presented. These trials consisted of a succession of congruent and incongruent combination of arrows. Using a two-vector method that included both reaction time and accuracy, a final score was calculated for school children who reached a high level of accuracy (>80%), which was as follows: (0.25 x number of correct responses) + $5 - \log_{10}$ [(congruent reaction time + incongruent reaction time)/ 2]. A total score considering accuracy was determined for children who scored less than 80%.

**Cognitive flexibility,** using the Dimension Change Card Sort (DCCS) [32]. Participants were shown two target cards and asked to order a group of bivalent test cards, first according to one dimension and then according to the other dimension. After a four-trial practice, children were given a 30-trial block with both 'shape' and 'color' requirements. Using reaction time and accuracy percentage on preswitch and postswitch, a raw score was determined. Using a two-vector method that integrated both accuracy and reaction time, a final score was calculated for children who reached a high level of accuracy (>80%) as follows: (0.167 x number of correct responses) + $5 - \log_{10}$ [(congruent reaction time + incongruent reaction time)/ 2]. For participants scoring <80%, a total outcome considering accuracy was determined.

**Working memory,** using the List Sorting Working Memory Test [33]. Children were given a group of pictures, and items were presented both auditorily and visually. Then, they were required to repeat the names of the items observed in order of size, from smallest to biggest. The number of items shown in each trial increased by one for each series. This test was made up of two parts. In the first one, all the items belonged to the same category (animals or food items). In the second one, the items from both categories (food items and animals) were presented together, and children had to repeat the items by category and size. The List Sorting "Total Score" was composed of final scores based upon a sum of the total correct trials across the two lists.

An EF Total Score, defined as the sum of the z-scores for inhibitory control, cognitive flexibility, and working memory, was calculated following the same standardization procedure described previously for GMC.

## Academic achievement

The final grades in language and mathematics, provided by the school administration, were used to determine AA. These subjects were selected because, in Spain, they are considered instrumental subjects, as they serve as a foundation for learning other subjects and help in everyday life and activities. The children's final grades from the previous academic year were used (range from 1 to 10 score, with 10 being the best grade). Since schools within the same region follow a standardized curriculum, assessment system, and grading criteria, these grades are comparable across students and schools. For the analyses, the arithmetic mean of language and mathematics grades was calculated, representing each student's academic performance over a complete school year.

## Anthropometry

Weight was measured using a scale (Seca 861, Vogel and Halke, Hamburg, Germany) with the child in light clothing and barefoot. For height, a wall stadiometer (Seca 222, Vogel and Halke) was used with children barefoot and standing upright with their sagittal midline in contact with the backboard. Both weight and height were measured twice, and their arithmetic mean was considered for the analysis. Body mass index (BMI) was calculated as weight (kg)/ height (m$^2$).

## Sociodemographic data

The age, sex and mother's education level were gathered using a questionnaire administered to the parents. Data on mother's education level as the maximum level of education attained by the children's mothers gathered the validated scale proposed by the Spanish Society of Epidemiology [34] to measure socioeconomic status. This questionnaire includes an item related to mother's education with six response options: i) no literacy skills; ii) no studies; iii) elementary studies; iv) secondary studies; v) high school; and vi) university studies. In our study, these six categories were collapsed into three, due to the small number of individuals in the lower and upper categories: lower/lower middle (no literacy skills, no studies, and elementary studies), middle (secondary studies) and upper middle/upper (high school and university studies). The mother's education level was used as covariate in the mediation models (as an interval variable), because it has been shown to be a strong predictor of children's AA [35].

## Statistical analysis

Interval variables were summarized as means and standard deviations. The mother's education level was expressed as counts and frequencies. Both statistical (Kolmogorov-Smirnov test) and graphical procedures (normal probability plots) were used to evaluate the goodness of fit of variables to a normal distribution and all variables fitted to a normal distribution. Differences in all variables between boys and girls were tested using Student's t-tests. Effect sizes were estimated using Cohen's *d*, with values below 0.20 considered negligible, 0.21–0.49 small, 0.50–0.79 medium, and ≥0.80 large [36].

Pearson correlation coefficients were estimated to examine the relationship between GMC, CRF, S/A, UBS, LBS, EF Total Score, and AA for the total sample and by sex. Values below 0.1 were defined as trivial, weak in the range 0.1–0.29, moderate in the interval 0.3–0.49, and large when greater than 0.5 [36]. A Fisher's Z-transformation test was performed to assess differences by sex, which were statistically significant [37].

The mediation effect of different fitness components (CRF, S/A, UBS, and LBS) and EF on the relationship between GMC and AA for the total sample and by sex was tested using the PROCESS SPSS Macro, version 4.2, selecting model 6 (serial multiple mediation analysis) and 10.000 bias-corrected bootstrap samples [38]. For this analysis, GMC was entered in the model as the independent variable, the proposed mediators were, consecutively, fitness variables and EF, and AA was included as the dependent variable. Each fitness component was individually entered in a different model: model a, CRF; model b, S/A; model c, UBS; and model d, LBS. In the mediation model, the total (c), and direct effects ($a_1$, $a_2$, $b_1$, $b_2$, $d_{12}$ and c´) were estimated. Additionally, this model examines three indirect effects (IEs) (fitness pathway,

cognitive pathway, and multiple pathway) that indicate the change in AA for each unit change in GMC that is mediated by each fitness variable and EF. The IEs were considered significant when the 95% confidence interval did not include zero.

The IE is represented by the pathway from GMC to AA via mediators (pathways a, b, and d). The fitness pathway ($a_1$-$b_1$), from GMC to AA via fitness variables (a, CRF; b, S/A; c, UBS; and d, LBS). The cognitive pathway ($a_2$-$b_2$), from GMC to AA via EF. Finally, the multiple pathway ($d_{12}$), from GMC to AA via fitness variables (a, CRF; b: S/A; c, UBS; and d, LBS) and EF. Proposed cut points to quantify effect size were 0.14, 0.36, and 0.51 for small, medium and large effect sizes, respectively [39]. The percentages of mediation ($P_M$) were calculated as (IE/total effect) x 100 to estimate the percentage of the total effect explained by the mediation pathways. All analyses were adjusted by age, BMI, and mother's education level.

Data analyses were conducted using IBM SPSS Statistics version 28.0 (IBM® SPSS®, Armonk, NY: IBM Corp.), and the level of significance was set at $p < 0.05$.

## Results

The characteristics of the total sample, as well as by sex, are presented in Table 1. Of the 562 schoolchildren involved in the study, 293 were girls (52.14%), with a mean age of 10.02 years (SD = ± 0.71). Boys scored significantly higher on aiming-catching ($p < 0.001$), CRF ($p < 0.001$), S/A ($p = 0.006$), UBS ($p = 0.004$), and LBS ($p < 0.001$) while girls scored significantly higher on static and dynamic balance ($p < 0.001$), and cognitive flexibility ($p = 0.014$).

The Pearson's correlation coefficients for the total sample, as well as by sex, are shown in Table 2. For the total sample, all the fitness variables showed a significant correlation with AA and EF (r values between 0.131 and 0.249, $p < 0.01$), except between UBS and EF. By sex, all the fitness variables show a significant correlation with AA and EF except between UBS and EF for both sexes, between UBS and AA for girls, and between LBS and AA for boys. In all the fitness variables, the correlation coefficient with both EF and AA was consistently higher in girls compared to boys. Specifically, S/A stands out as the fitness variable with the strongest correlation with EF (r = −0.217 in boys and r = −0.269 in girls; $p < 0.01$), while CRF exhibits the highest correlation with AA (r = 0.219 in boys and r = 0.317 in girls; $p < 0.01$). Most of the mentioned associations show weak correlation strength. However, notable moderate strength correlations are observed between CRF and AA (r = 0.317; $p < 0.01$), as well as between LBS and EF (r = 0.312; $p < 0.01$), both exclusively in girls.

Multiple mediation models for the total sample indicated a consistent association between GMC and AA across all models, with unstandardized coefficients ranging from 0.203 to 0.205 ($p < 0.001$). Most of this effect was explained by the direct association, with coefficients between 0.105 and 0.149 ($p < 0.05$). Additionally, all three indirect pathways (fitness, cognitive, and multiple) were significant for all fitness variables, except for the fitness pathway in UBS and LBS, and the multiple pathway in S/A and UBS (Fig 1).

Among boys (Fig 2) the total effect of GMC on AA was significantly lower, with coefficients ranging from 0.119 to 0.130 ($p < 0.05$) highlighting no direct association between GMC and AA for any of the variables studied. Instead, improvements in AA through GMC consistently occurred via EF (cognitive pathway). In contrast, among girls (Fig 2), the total effects were significantly higher (0.295 to 0.304, $p < 0.001$), with most of the effect explained by the direct association between GMC and AA, which was significant across all models, with coefficients ranging from 0.200 to 0.252 ($p < 0.05$). Additionally, for CRF and S/A, both the fitness and multiple pathways were significant, while only the cognitive pathway was significant for UBS, and only the multiple pathway was significant for LBS.

The total, direct, and IEs of the different pathways from the serial multiple mediation analyses, along with the mediation percentages for direct and IEs, are summarized in Table 3. Across all models examined, a positive association between GMC and AA was observed. For the total sample, most of the total effect was accounted for the direct effect (51.22% to 72.68%), followed by the IEs of the cognitive pathway (19.51% to 30.73%), fitness pathway (10.84% to 19.02%), and multiple pathway (4.43% to 9.27%). In boys, more than half of the total effect was explained by the IE of the cognitive pathway (56.35% to 68.91%). In contrast, in girls, the majority of the total effect was attributed to the direct effect of GMC on

**Table 1. Characteristics of the study sample by sex.**

| | Total Sample (n = 562) | Boys (n = 269) | Girls (n = 293) | p-value | Cohen's *d* |
|---|---|---|---|---|---|
| Sample Characteristics | | | | | |
| Age (years) | 10.02 (0.71) | 10.01 (0.71) | 10.04 (0.71) | 0.291 | 0.042 |
| Weight (kg) | 36.55 (9.77) | 36.62 (9.72) | 36.48 (9.84) | 0.433 | 0.014 |
| Height (cm) | 140.91 (7.33) | 140.99 (7.07) | 140.84 (7.56) | 0.407 | 0.020 |
| Body Mass Index (kg/m$^2$) | 18.22 (3.78) | 18.25 (3.86) | 18.19 (3.72) | 0.423 | 0.016 |
| Mother's Education Level [n (%)] | | | | | |
| Lower/lower middle | 62 (12.38) | 28 (11.76) | 34 (12.93) | 0.641 | |
| Middle | 221 (44.11) | 102 (42.86) | 119 (45.25) | 0.492 | |
| Upper middle/upper | 218 (43.51) | 108 (45.38) | 110 (41.82) | 0.548 | |
| Gross Motor Competence (GMC) | | | | | |
| Aiming-Catching (n) | 7.13 (1.78) | 7.76 (1.67) | 6.55 (1.68) | **< 0.001** | 0.722 |
| Static Balance (s) | 18.47 (8.97) | 16.80 (8.87) | 20.02 (8.80) | **< 0.001** | 0.364 |
| Dynamic Balance (p) | 9.30 (1.68) | 8.99 (1.99) | 9.58 (1.27) | **< 0.001** | 0.353 |
| GMC Total Score$^Z$ | 0.3880 (1.89) | 0.4123 (2.00) | 0.3656 (1.79) | 0.385 | 0.025 |
| Fitness | | | | | |
| CRF (VO$_2$max) | 45.97 (4.61) | 47.41 (4.98) | 44.66 (3.81) | **< 0.001** | 0.665 |
| Speed/Agility (s)* | 13.71 (1.36) | 13.56 (1.40) | 13.85 (1.32) | **0.006** | 0.213 |
| Upper Body Strength (kg) | 12.39 (3.49) | 12.80 (3.77) | 12.01 (3.17) | **0.004** | 0.227 |
| Lower Body Strength (cm) | 116.66 (20.71) | 121.55 (21.41) | 112.23 (19.04) | **< 0.001** | 0.460 |
| Executive Function (EF) | | | | | |
| Inhibitory Control (FT)$^¥$ | 19.66 (1.35) | 19.61 (1.36) | 19.70 (1.33) | 0.172 | 0.067 |
| Cognitive Flexibility (DCCS)$^§$ | 28.17 (3.10) | 27.87 (3.10) | 28.44 (3.08) | **0.014** | 0.184 |
| Working Memory (LSWM) | 14.25 (3.18) | 14.45 (3.31) | 14.06 (3.04) | 0.077 | 0.123 |
| EF Total Score$^Z$ | 0.0037 (1.91) | −0.0030 (1.96) | 0.0099 (1.88) | 0.469 | 0.007 |
| Academic Achievement (AA) | | | | | |
| Language-Mathematics$^Δ$ | 7.04 (1.73) | 7.01 (1.74) | 7.07 (1.73) | 0.336 | 0.035 |

Values are presented as mean ± standard deviation, except for mother's education level, which are shown as number and percentages [n (%)].

Abbreviations. GMC, gross motor competence; CRF, cardiorespiratory fitness; EF, executive function; FT, flanker task; DCCS, dimensional change card sort test; LSWM, list shorting working memory test. Test scores: n, number of right attempts; s, seconds; p, punctuation.

In bold (p-values), statistical signification (p < 0.05).

Cohen's *d* effect sizes (Cohen, 1988): < 0.20 no effect; 0.21–0.49, small effect; 0.50–0.79 medium effect.

Higher scores indicate better performance, except in Speed/Agility*, where lower scores (shorter times) indicate better performance.

Z GMC Total score was calculated from z scores of aiming-catching, static balance and dynamic balance.

Z EF Total score was calculated from z total scores of inhibitory control, cognitive flexibility and working memory.

¥ Total score was calculated using a two-vector method that incorporates both accuracy and reaction time. Total score = (0.25 x number correct responses) + 5 − $\log_{10}$ [(congruent reaction time + incongruent reaction time)/2].

§ Total score was calculated using a two-vector method that incorporates both accuracy and reaction time. Total score = (0.167 x number correct responses) + 5 − $\log_{10}$ [(congruent reaction time + incongruent reaction time)/2].

$^Δ$Mean of the grades in language and mathematics.

AA (66.45% to 85.42%), followed by the IEs of the fitness pathway (19.08% to 20.27%), cognitive pathway (for UBS only, 14.92%), and multiple pathway (4.93% to 9.43%).

## Discussion

This study is, to our knowledge, the first to apply multiple mediation models to explore how physical fitness and EF influence the link between GMC and AA in children. Our findings show that this relationship is partially explained through three

**Table 2. Bivariate correlation coefficients among the study variables for the total sample and by sex.**

| | | GMC | CRF | Speed/ Agility | Upper Body Strength | Lower Body Strength | Executive Function Total Score |
|---|---|---|---|---|---|---|---|
| CRF | Total | 0.278** | | | | | |
| | Boys | 0.270** | | | | | |
| | Girls | 0.313** | | | | | |
| Speed/Agility^T | Total | −0.410** | −0.490** | | | | |
| | Boys | −0.441** | −0.487** | | | | |
| | Girls | −0.380** | −0.483** | | | | |
| Upper Body Strength | Total | 0.127** | 0.003 | −0.138** | | | |
| | Boys | 0.185** | 0.013 | −0.204**^ | | | |
| | Girls | 0.055 | −0.095 | −0.042^ | | | |
| Lower Body Strength | Total | 0.333** | 0.469** | −0.544** | 0.214** | | |
| | Boys | 0.354** | 0.497** | −0.620** | 0.311**^^ | | |
| | Girls | 0.367** | 0.473** | −0.556** | 0.144*^^ | | |
| Executive Function Total Score | Total | 0.229** | 0.181** | −0.241** | 0.064 | 0.249** | |
| | Boys | 0.262** | 0.161* | −0.217** | 0.049 | 0.177**^ | |
| | Girls | 0.194** | 0.233** | −0.269** | 0.083 | 0.312**^ | |
| Academic Achievement^Δ | Total | 0.225** | 0.246** | −0.237** | −0.113** | 0.131** | 0.417** |
| | Boys | 0.171** | 0.219** | −0.201** | −0.142* | 0.061^ | 0.506**^^ |
| | Girls | 0.282** | 0.317** | −0.278** | −0.079 | 0.209**^ | 0.331**^^ |

Higher coefficients indicate stronger correlations, except for Speed/Agility^T, where lower coefficients reflect stronger performance (inverse variable).

Interpretation of correlation coefficients based on Cohen (1988): 0.10–0.29, small; 0.30–0.49, medium; > 0.50, large.

^Δ Mean of the grades in language and mathematics.

Abbreviations: GMC, Gross motor competence; CRF, Cardiorespiratory fitness.

Fisher Z-Transformation indicate statistically significant differences by sex: ^p < 0.05; ^^p < 0.01.

*p < 0.05; **p < 0.01.

distinct pathways: a fitness pathway (especially via CRF and S/A), a cognitive pathway (via EF), and a multiple pathway involving both fitness and cognition. These pathways remained significant even after adjusting for age, BMI, and maternal education level. Importantly, we observed sex-specific patterns: boys showed exclusive mediation through cognition, while girls showed several mediation pathways, including fitness and cognitive components.

These results align with prior studies. For example, Schmidt et al. [10] found that EF fully mediated the GMC–AA link in children using structural equation modeling, and our earlier work also highlighted the partial mediating role of EF [12]. Cadoret et al. [5] similarly observed cognitive mediation between motor proficiency and academic performance. However, only one previous study [14] specifically tested CRF as a mediator, finding no mediation—contrary to our results. This discrepancy could be explained by the different instruments and predictive methodologies used to evaluate CRF, MC, and AA, as well as changes in educational level and the transformations that occur during puberty.

Our results suggest that improving GMC may enhance physical fitness and cognitive abilities, ultimately benefiting academic outcomes. This was especially evident in girls, where CRF, S/A, and LBS contributed to the mediation process. These findings underscore the importance of designing motor interventions that enhance both fitness (particularly CRF and S/A) and EF, especially in educational settings targeting children aged 9–11.

Our study revealed notable sex differences in the relationship between GMC and AA. Among girls, this relationship appeared more complex, involving both direct associations and multiple mediating pathways through fitness (CRF and S/A), cognition (UBS), and combined effects (CRF, S/A, and LBS). In contrast, for boys, the relationship was mediated only through cognition, aligning with findings by Fernández-Sánchez et al. [12], but differing from Lopes et al. [11], who

## a) Cardiorespiratory Fitness (CRF)

## b) Speed/Agility (S/A)

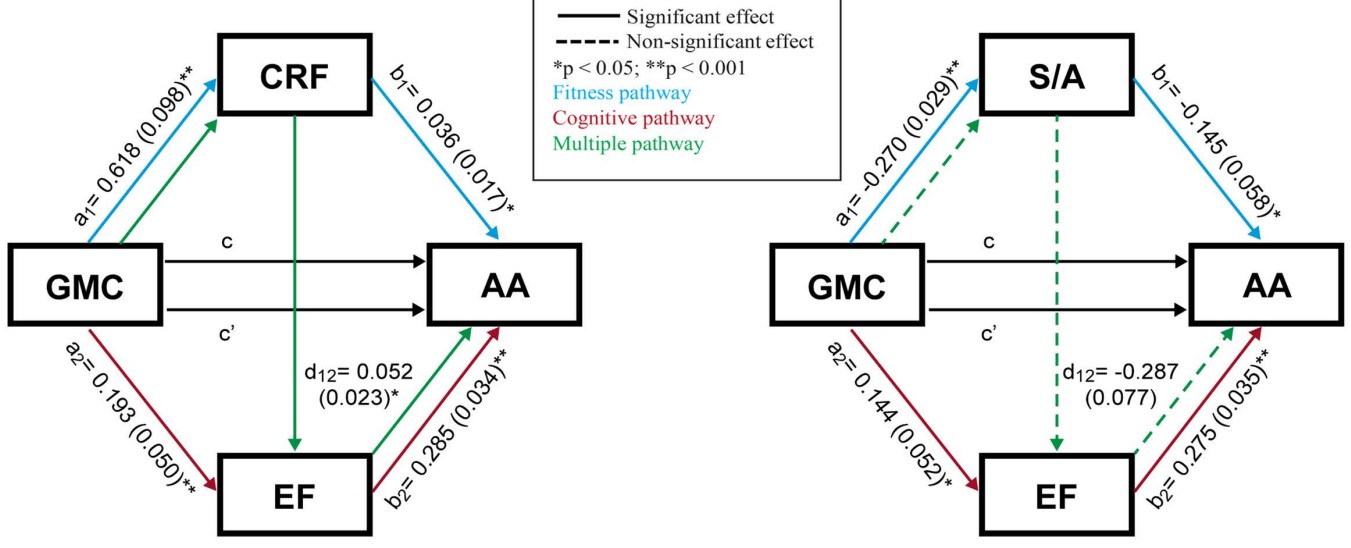

## c) Upper Body Strength (UBS)

## d) Lower Body Strength (LBS)

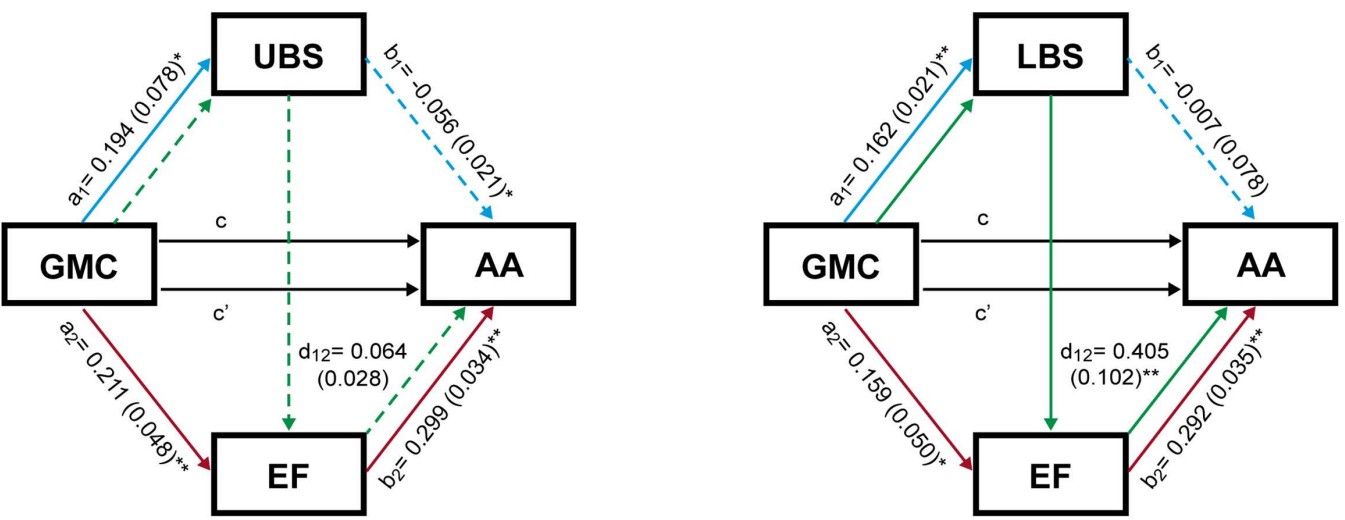

**Fig 1. Serial multiple mediation model (total sample).** Association between gross motor competence (independent variable) and academic achievement (dependent variable), with different fitness components (CRF, S/A, UBS and LBS) and executive function as mediators, controlling for age, body mass index and mother's education level. Values for the $a_1$, $a_2$, $b_1$, $b_2$ and $d_{12}$ pathways are expressed as the unstandardized regression coefficients (standard error). Values c and c' express total and direct effect, respectively. GMC, gross motor competence; CRF, cardiorespiratory fitness; S/A, speed/agility; UBS, upper body strength; LBS, lower body strength; EF, executive function; AA academic achievement.

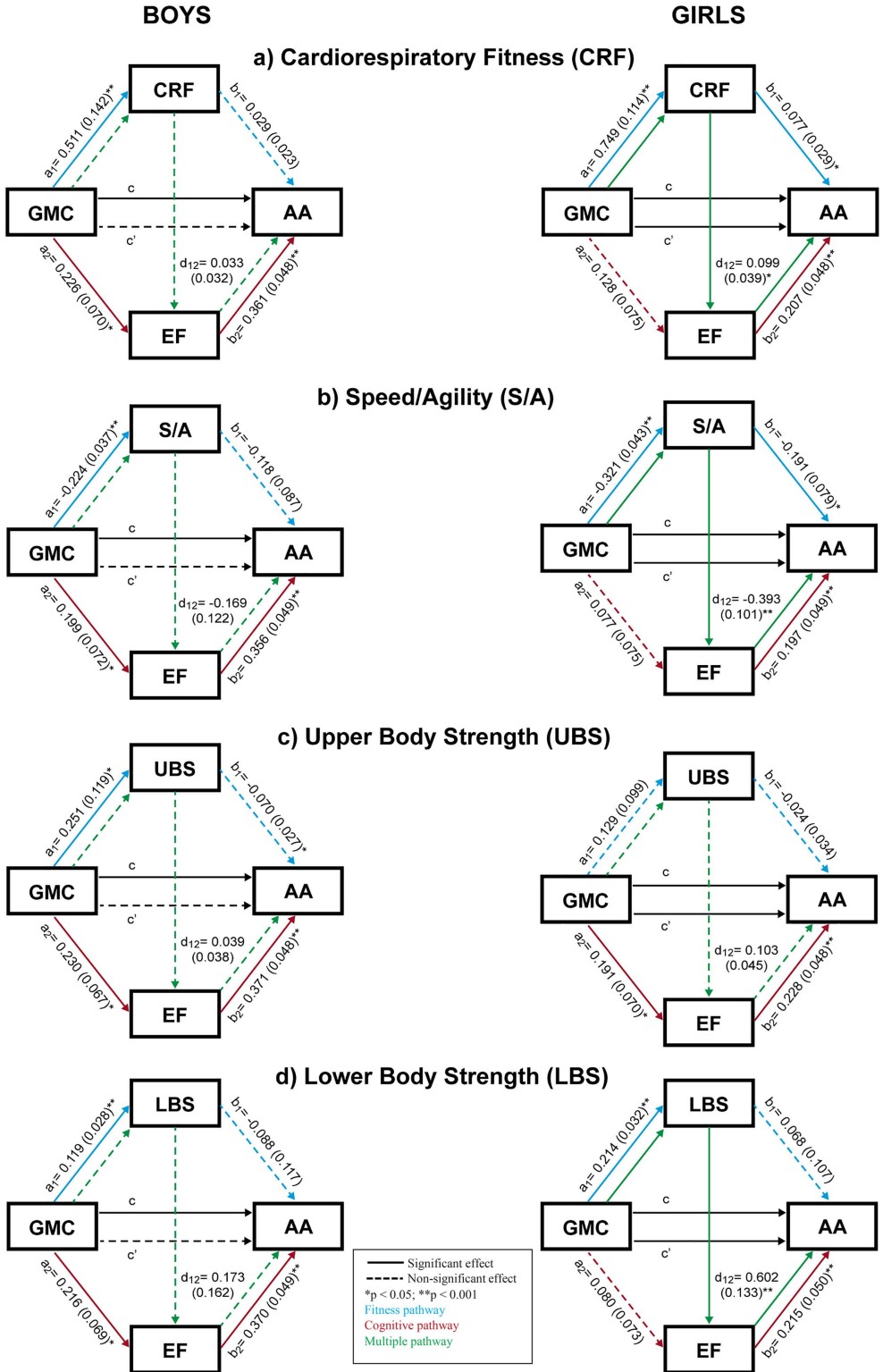

**Fig 2. Serial multiple mediation model (boys and girls).** Association between gross motor competence (independent variable) and academic achievement (dependent variable), with different fitness components (CRF, S/A, UBS and LBS) and executive function as mediators, controlling for age, body mass index and mother's education level. Values for the $a_1$, $a_2$, $b_1$, $b_2$ and $d_{12}$ pathways are expressed as the unstandardized regression coefficients

(standard error). Values c and c' express total and direct effect, respectively. GMC, gross motor competence; CRF, cardiorespiratory fitness; S/A, speed/agility; UBS, upper body strength; LBS, lower body strength; EF, executive function; AA academic achievement.

**Table 3. Total, direct, and indirect effects of the different pathways from the serial multiple mediation analyses, investigating fitness and executive function as mediators between gross motor competence and academic achievement, controlling for age, body mass index, and mother's education level.**

| | Fitness Mediator Variable | Total Effect (c) | Direct Effect (c') | | Indirect Effects | | | | | | | | | | | |
| | | | | | FITNESS PATHWAY | | | | COGNITIVE PATHWAY | | | | MULTIPLE PATHWAY | | | |
| | | | Direct Effect (c') | P_M (%) | Indirect Effect | 95% CI | | P_M (%) | Indirect Effect | 95% CI | | P_M (%) | Indirect Effect | 95% CI | | P_M (%) |
| | | | | | | Lower | Upper | | | Lower | Upper | | | Lower | Upper | |
| Total | CRF | 0.203 (0.038)** | 0.117 (0.038)* | 57.64 | 0.022 (0.012)* | 0.001 | 0.047 | 10.84 | 0.055 (0.017)** | 0.023 | 0.088 | 27.09 | 0.009 (0.005)* | 0.001 | 0.02 | 4.43 |
| | S/A | 0.205 (0.038)** | 0.105 (0.039)* | 51.22 | 0.039 (0.016)* | 0.008 | 0.072 | 19.02 | 0.040 (0.016)* | 0.009 | 0.073 | 19.51 | | | | |
| | UBS | 0.205 (0.038)** | 0.149 (0.036)** | 72.68 | | | | | 0.063 (0.016)** | 0.032 | 0.097 | 30.73 | | | | |
| | LBS | 0.205 (0.038)** | 0.141 (0.038)** | 68.78 | | | | | 0.046 (0.016)* | 0.016 | 0.079 | 22.44 | 0.019 (0.006)** | 0.009 | 0.033 | 9.27 |
| Boys | CRF | 0.119 (0.054)* | | | | | | | 0.082 (0.027)* | 0.029 | 0.137 | 68.91 | | | | |
| | S/A | 0.126 (0.054)* | | | | | | | 0.071 (0.029)* | 0.018 | 0.130 | 56.35 | | | | |
| | UBS | 0.131 (0.053)* | | | | | | | 0.085 (0.027)* | 0.035 | 0.141 | 64.89 | | | | |
| | LBS | 0.130 (0.054)* | | | | | | | 0.080 (0.027)* | 0.030 | 0.135 | 61.54 | | | | |
| Girls | CRF | 0.304 (0.054)** | 0.204 (0.056)** | 67.11 | 0.058 (0.027)* | 0.008 | 0.114 | 19.08 | | | | | 0.015 (0.008)* | 0.003 | 0.035 | 4.93 |
| | S/A | 0.301 (0.054)** | 0.200 (0.057)* | 66.45 | 0.061 (0.025)* | 0.015 | 0.115 | 20.27 | | | | | 0.025 (0.010)** | 0.008 | 0.047 | 8.31 |
| | UBS | 0.295 (0.055)** | 0.252 (0.053)** | 85.42 | | | | | 0.044 (0.020)* | 0.009 | 0.088 | 14.92 | | | | |
| | LBS | 0.297 (0.054)** | 0.237 (0.057)** | 79.80 | | | | | | | | | 0.028 (0.010)** | 0.011 | 0.05 | 9.43 |

Results showed unstandardized coefficients (standard error) and bootstraps confidence 95% confidence interval (CI) on 10000 bootstraps.

CRF, cardiorespiratory fitness; S/A, speed/agility; UBS, upper body strength; LBS, lower body strength; $P_M$, percentage of mediation.

Interpretation of effect sizes based on Cheung (2009): 0.14–0.35, small; 0.36–0.50, medium; >0.51, large.

*p<0.05; **p<0.001.

reported no significant sex differences. The presence of both direct and fitness-mediated associations in girls suggests a more integrated development of physical and cognitive domains compared to boys. These disparities may be explained by biological and sociocultural factors influencing motor and cognitive development [40], such as sex-specific patterns of biological maturation [41], brain development [42], and gender-related activity preferences [43]. Another possible explanation is that boys and girls may engage distinct neural circuits and molecular mechanisms when addressing the same cognitive challenges, leading to differences in strategy despite comparable overall ability [44]. Given the limited evidence focused specifically on sex differences, further research is needed.

When comparing the strength of the mediation pathways, we found that in the total sample, the cognitive pathway was the most influential, followed by the fitness pathway and the multiple pathway. For boys, cognition was the sole mediator, supporting the "cognitive stimulation hypothesis" [45,46]. In girls, however, the fitness pathway—particularly through CRF and S/A—was the dominant route, aligning more closely with the "cardiovascular fitness hypothesis" [47]. These differences may reflect underlying neurodevelopmental trajectories, particularly around age 9–10, when sex differences in brain maturation are especially pronounced [48].

Finally, our findings underscore the role of CRF and S/A as key mediators of cognitive benefits, whereas the contribution of muscular strength remains less clear. This distinction may reflect different biological and neurological mechanisms. CRF has been linked to angiogenesis in the motor cortex and increased cerebral blood flow, improving brain vascularization and supporting cognitive performance [49]. Similarly, S/A may enhance attention [50] and accelerate neural impulse conduction, thereby increasing processing speed [16]. These adaptations, together with enhanced spinal cord function, synaptogenesis, greater BDNF release, and reorganization of motor cortex representations [51], likely contribute to improved cognition [52] and, ultimately, to better AA [16]. However, the role of muscular strength, particularly UBS and LBS, remains uncertain. Evidence regarding its association with cognition and AA is mixed: while some studies report positive correlations [52,53], others find no significant relationship [54,55]. Thus, further research is required to clarify the contribution of muscular strength, and more broadly of physical fitness components, to cognitive and academic outcomes.

## Limitations

This study has several limitations. First, its cross-sectional design limits causal inference. Longitudinal studies are needed to confirm the temporal direction of these relationships. Nevertheless, the temporality proposed in our models is consistent with the current knowledge about the relationship between GMC, fitness, EF, and AA [5,10,12,56,57]. Second, while school-based academic assessments are useful, they may not reflect the full range of cognitive abilities or academic skills. Third, we did not account for behavior-related factors such as physical activity or sleep, both of which can influence fitness, cognition, and academic outcomes. Future studies should objectively measure physical activity (e.g., using accelerometers) and sleep quality (e.g., actigraphy or parent-reported sleep diaries) to better understand their potential confounding or mediating roles. Finally, variables such as visual acuity or adiposity were not considered, though both may affect motor test performance [58–60].

## Conclusions

Our findings indicate that the association between GMC and AA in children is largely explained by improvements in CRF, S/A, and EF, with distinct mediation patterns by sex. These results emphasize the importance of implementing school-based programs that integrate motor skill development with both fitness and cognitive training. Specifically, boys may benefit more from cognitively demanding, perceptual–motor activities, while girls may respond better to interventions prioritizing physical fitness enhancement.

From a practical perspective, educators and policymakers should prioritize daily opportunities for structured physical activity that go beyond regular physical education classes, incorporating aerobic, coordinative, and cognitively engaging tasks. Such approaches could foster academic performance, reduce educational inequalities, and contribute to healthier developmental trajectories.

Future research should examine the long-term effects of integrated motor–cognitive interventions on AA, explore the neurobiological mechanisms underlying these associations, and consider additional mediators such as sleep, attention regulation, and motivation. By framing MC not only as a physical education outcome but as a multidisciplinary strategy, schools can promote cognitive development, academic success, and lifelong health.

## Acknowledgments

The authors thank the schools, families, and children for their enthusiastic participation in the study. We thank all membership of the Cuenca Study Group, who helped to make this study possible: Carlos Berlanga-Macías, Blanca Notario-Pacheco, María Jesús Pardo-Guijarro, Celia Álvarez-Bueno, Marta Nieto-López, Alberto González-García, Jorge Cañete García-Prieto, Ana Torres-Costoso, Antonio García-Hermoso, Caterina Pesce, and Ricardo Cuevas-Campos.

## Author contributions

**Conceptualization:** Antonio Fernández-Sánchez, Mairena Sánchez-López.

**Data curation:** Mairena Sánchez-López.

**Funding acquisition:** Vicente Martínez-Vizcaíno.

**Methodology:** Antonio Fernández-Sánchez, Vicente Martínez-Vizcaíno.

**Resources:** Vicente Martínez-Vizcaíno.

**Supervision:** Abel Ruiz-Hermosa, Andrés Redondo-Tébar, Ana Díez-Fernández, Vicente Martínez-Vizcaíno, María Eugenia Visier-Alfonso, Mairena Sánchez-López.

**Validation:** Abel Ruiz-Hermosa, Andrés Redondo-Tébar, Ana Díez-Fernández, Vicente Martínez-Vizcaíno, Mairena Sánchez-López.

**Visualization:** María Eugenia Visier-Alfonso, Mairena Sánchez-López.

**Writing – original draft:** Antonio Fernández-Sánchez.

**Writing – review & editing:** Abel Ruiz-Hermosa, Andrés Redondo-Tébar, Vicente Martínez-Vizcaíno, Mairena Sánchez-López.

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
