## [Decision Letter · Decision Letter 0]

17 Jul 2025

Relation between motor competence and academic achievement: the mediating role of fitness and cognition in boys and girls

PLOS ONE

Dear Dr. Ruiz-Hermosa,

Thank you for submitting your manuscript to PLOS ONE. After careful consideration, we feel that it has merit but does not fully meet PLOS ONE’s publication criteria as it currently stands. Therefore, we invite you to submit a revised version of the manuscript that addresses the points raised during the review process.

We look forward to receiving your revised manuscript.

Kind regards,

Helmi Chaabène

Academic Editor

PLOS ONE

Journal Requirements:

2. In the ethics statement in the Methods, you have specified that verbal consent was obtained. Please provide additional details regarding how this consent was documented and witnessed, and state whether this was approved by the IRB.

“The Ministry of Economy and Competitiveness-Carlos III Health Institute (FIS PI16/01919) funded this study. Additional funding was obtained from the Research Network on Preventative Activities and Health Promotion (RD12/0005/0009). Furthermore, Andrés Redondo-Tébar and Abel Ruiz-Hermosa are postdoctoral researchers funded by the Margarita Salas Fellowship through the University of Castilla-La Mancha “Next Generation EU” (2022-POST-21124 and 2021-MS-20547, respectively).”

5. In the online submission form, you indicated that “The data underlying the results presented in the study are available upon reasonable request to the corresponding author”

6. PLOS requires an ORCID iD for the corresponding author in Editorial Manager on papers submitted after December 6th, 2016. Please ensure that you have an ORCID iD and that it is validated in Editorial Manager. To do this, go to ‘Update my Information’ (in the upper left-hand corner of the main menu), and click on the Fetch/Validate link next to the ORCID field. This will take you to the ORCID site and allow you to create a new iD or authenticate a pre-existing iD in Editorial Manager.

Additional Editor Comments:

Dear authors, 

I'm sorry for the delay in providing our feedback. We were in the process of seeking additional reviewers after identifying a potential conflict of interest between at least one of the initial reviewers and the research team. Unfortunately, we were not very successful in recruiting new reviewers. To avoid any further delay, I have reviewed the manuscript myself. In my assessment, the study is well designed and executed. The topic is both interesting and relevant, and to the best of my knowledge, it has not been previously addressed, making it a novel contribution. The sample size is appropriate, and the main findings are compelling. They lay a strong foundation for future research to replicate the results and explore the potential role of sex as a moderating variable.

Please make sure to address all reviewers’ comments and suggestions in a point-by-point response and implement the corresponding amendments to the manuscript where appropriate.

I'm looking forward to receive the revised version. 

All the best,

Helmi Chaabene

Reviewers' comments:

Reviewer's Responses to Questions

**Comments to the Author**

1. Is the manuscript technically sound, and do the data support the conclusions?

Reviewer #1: Yes

Reviewer #2: Yes

Reviewer #3: Yes

Reviewer #4: Yes

2. Has the statistical analysis been performed appropriately and rigorously?

Reviewer #1: Yes

Reviewer #2: Yes

Reviewer #3: Yes

Reviewer #4: Yes

3. Have the authors made all data underlying the findings in their manuscript fully available?

Reviewer #1: Yes

Reviewer #2: Yes

Reviewer #3: Yes

Reviewer #4: Yes

4. Is the manuscript presented in an intelligible fashion and written in standard English?

Reviewer #1: Yes

Reviewer #2: Yes

Reviewer #3: Yes

Reviewer #4: Yes

Reviewer #1: I would like to thank authors for this interesting study. Academic achievement is an important subject and it has a relation with physical fitness. I believe this study has many strengths. First, it has a very good sample size and tests and measurements are appropriate. In addition, it has a strong results and discussion sections. However, authors need to report reliability and validity of tests and measurements. In addition, this study needs a stronger conclusion section with future perspectives and practical applications. Lastly, this study needs more up to date (2022 and above literature review. I look forward to seeing edited version of this manuscript. Best regards.

Reviewer #2: Este artículo entrega respuestas a una serie de inquietudes que se presentan en el campo de la educación. La indagación tiene un correcto desarrollo que ayuda a ampliar el conocimiento respecto a la temática en cuestión. Por tanto, el presente artículo cumple correctamente las etapas en cuanto a forma y fondo.

Reviewer #3: REVIEW REPORT (PONE-D-24-52528)

Relation between motor competence and academic achievement: the mediating role of fitness and cognition in boys and girls

ABSTRACT

- Please adjust the acronym “AA” that was used without definition and review the presentation of the percentages and effect coefficients for adequate contextualization, focusing more on the main findings in interpretative language (for example, “cognitive mediation was more important in boys, while fitness was more expressive in girls”).

INTRODUCTION

- Overall, it is ok. But I believe that the hypothesis needs to be more evident, such as “it is expected that executive functions and fitness components partially mediate the relationship between motor competence and academic performance, with variations according to sex”.

- How does the present study differ from previous studies reporting the absence or presence of mediation? Methodological inconsistencies? Different samples or measures of physical fitness?

METHODS

- Were only students with complete data included? Were students with disabilities excluded? Please clarify.

- I recommend a better explanation of the z-score procedure for GMC and EF. Is it a standardization within the sample? Between age groups?

- Was there standardization of grades among schools, that is, were the evaluation criteria uniform? Please include this information.

RESULTS

- I really believe that tables 1 and 2 should contain clearer notes about the units and meaning of the tests, such as "the higher, the better", or "shorter time = better agility". In addition, use the magnitude of the effects explicitly if the effect is small, medium or large as suggested by Cohen, especially in indirect effects.

- The results are ok, with no evidence of artificial homogeneity or variance suppression, but ambiguity about causality is needed and to avoid undue temporal inference in a cross-sectional study. Please review this throughout the text.

DISCUSSION

- It is well written, but, in general, the authors should condense the paragraphs, avoiding repeating previous numerical results.

- The authors should moderate the first paragraph to use more accessible language instead of excessively technical. The text should convey the main message to all readers and not just the specialized ones. For example, boys showed exclusive cognitive mediation, while girls had multiple mediation pathways.

- The authors mention the general results and highlight the importance of the cognitive pathway in boys, but it is not clear why the physical components UBS and LBS did not act as mediators for them, contrary to what occurred in part with girls. Please review this, albeit objectively.

- How can PE teachers or policy makers use this evidence? This is not clear.

- The authors also inform the limitation of not measuring physical activity or sleep. Objectively, I suggest including objective information with suggestions on how to overcome this in future studies.

Reviewer #4: I appreciate the opportunity to review this manuscript titled “Relation between motor competence and academic achievement: The mediating role of fitness and cognition in boys and girls”. In this study, the authors assess the mediating role of fitness and executive functions on the motor competence and academic achievement relationship. I think that a major revision is necessary before accepting it for publication.

ABSTRACT

Although, in general, I consider that the abstract is well synthesized and provides the necessary information, it would be advisable to specify and/or define what is meant by multiple path in the results section. It is explained in the full text of the article, but if you have not read it beforehand, this aspect is not clear in the abstract..

INTRODUCTION

The introduction is well-written and the objectives are clearly stated. In this respect, no suggestions for improvement are made.

METHODS

The methodology used in the research is well explained and appropriate to achieve the objectives of the work.

RESULTS

- Referring to Table 1, it would be appropriate to add the effect size (Cohen's d) of the t-tests.

- The authors show in Table 2 the total and group-wise correlations for the main study variables. In relation to these results, it would perhaps be interesting to also add a statistical test (e.g. Fisher's Z-transformation) that allows to know concretely which specific correlations are statistically significantly different according to gender.

DISCUSSION

- The discussion is well-written. In this respect, no suggestions for improvement are made.

**Do you want your identity to be public for this peer review?** For information about this choice, including consent withdrawal, please see our Privacy Policy

Reviewer #1: **Yes: ** Ferman Konukman

Reviewer #2: No

Reviewer #3: No

Reviewer #4: No

---

## [Author Response · Author response to Decision Letter 1]

8 Sep 2025

Professor Helmi Chaabène

Please find enclosed a revised version of our manuscript “Relation between motor competence and academic achievement: the mediating role of fitness and cognition in boys and girls” (PONE-D-24-52528). We would like to thank you for giving us the opportunity to revise and improve our manuscript again; we also thank the reviewers for the thoughtful and constructive comments. We have considered the suggestions and have incorporated them into the revised manuscript, and as a result, we believe our manuscript is stronger. An itemized point-by-point response to the reviewers’ comments is shown below. The locations of the modifications made in the manuscript are detailed (page number, line number), as indicated in the file with track changes.

Thank you for your attention to this matter.

Abel Ruiz-Hermosa (corresponding author).

Social and Health Research Center. University of Castilla-La Mancha, Cuenca, Spain.

Journal Requirements:

1.Please ensure that your manuscript meets PLOS ONE's style requirements, including those for file naming. The PLOS ONE style templates can be found at:

Authors' response: Thank you very much for your feedback. We have reviewed the manuscript, and its format has been adjusted in accordance with the style templates provided.

2. In the ethics statement in the Methods, you have specified that verbal consent was obtained. Please provide additional details regarding how this consent was documented and witnessed, and state whether this was approved by the IRB.

Authors' response: Thank you very much for your comment. We have revised the relevant paragraph to provide a more detailed explanation of how verbal consent was obtained. The revised paragraph now reads as follows (page 8, line 187):

The study protocol was approved by the Clinical Research Ethics Committee of the ‘Virgen de la Luz’ Hospital in Cuenca (REG: 2016/PI021). Before data collection, an informational session was conducted in each classroom to explain the study’s objectives, the voluntary nature of participation, the anonymity of responses, and the right to withdraw at any time without consequences. These aspects were communicated in age-appropriate language to ensure the children’s understanding. Following this explanation, each child was explicitly asked to give their verbal consent, in accordance with the procedure approved by the ethics committee. This consent was given in the presence of the classroom teacher, the responsible researchers, and the children’s peers, who served as witnesses. In addition, written informed consent was obtained from the parents or legal guardians of all participants.

Authors' response: Thank you very much for your helpful correction. We have carefully revised the ‘Funding Information’ and ‘Financial Disclosure’ sections, and they now align accordingly.

“The Ministry of Economy and Competitiveness-Carlos III Health Institute (FIS PI16/01919) funded this study. Additional funding was obtained from the Research Network on Preventative Activities and Health Promotion (RD12/0005/0009). Furthermore, Andrés Redondo-Tébar and Abel Ruiz-Hermosa are postdoctoral researchers funded by the Margarita Salas Fellowship through the University of Castilla-La Mancha “Next Generation EU” (2022-POST-21124 and 2021-MS-20547, respectively).”

Authors' response: In ‘Financial disclosure’ section, we have stated that: "The funders had no role in study design, data collection and analysis, decision to publish, or preparation of the manuscript”. This information has also been incorporated into the cover letter.

5. In the online submission form, you indicated that “The data underlying the results presented in the study are available upon reasonable request to the corresponding author”.

Authors' response: We sincerely appreciate your comment. The minimal anonymized dataset required to replicate our study findings has been published in the ZENODO public repository and is openly accessible at the following URL: https://zenodo.org/records/17070143 (DOI: 10.5281/zenodo.17070143).

6. PLOS requires an ORCID iD for the corresponding author in Editorial Manager on papers submitted after December 6th, 2016. Please ensure that you have an ORCID iD and that it is validated in Editorial Manager. To do this, go to ‘Update my Information’ (in the upper left-hand corner of the main menu), and click on the Fetch/Validate link next to the ORCID field. This will take you to the ORCID site and allow you to create a new iD or authenticate a pre-existing iD in Editorial Manager.

Authors' response: Done.

Additional Editor Comments:

Dear authors,

I'm sorry for the delay in providing our feedback. We were in the process of seeking additional reviewers after identifying a potential conflict of interest between at least one of the initial reviewers and the research team. Unfortunately, we were not very successful in recruiting new reviewers. To avoid any further delay, I have reviewed the manuscript myself. In my assessment, the study is well designed and executed. The topic is both interesting and relevant, and to the best of my knowledge, it has not been previously addressed, making it a novel contribution. The sample size is appropriate, and the main findings are compelling. They lay a strong foundation for future research to replicate the results and explore the potential role of sex as a moderating variable.

Please make sure to address all reviewers’ comments and suggestions in a point-by-point response and implement the corresponding amendments to the manuscript where appropriate.

I'm looking forward to receive the revised version.

All the best,

Helmi Chaabène

Authors' response: Thank you very much for your feedback.

Response to Reviewer 1 Comments:

Reviewer #1: I would like to thank authors for this interesting study. Academic achievement is an important subject, and it has a relation with physical fitness. I believe this study has many strengths. First, it has a very good sample size, and tests and measurements are appropriate. In addition, it has strong results and discussion sections. However, authors need to report reliability and validity of tests and measurements. In addition, this study needs a stronger conclusion section with future perspectives and practical applications. Lastly, this study needs more up to date (2022 and above literature review. I look forward to seeing edited version of this manuscript. Best regards.

Authors' response: Thank you very much for your feedback. To report the reliability and validity of the tests and measurements, we have added the following sentences in the 'Instruments and Study Variables' section—specifically when defining each battery or tool—to provide evidence of their reliability and validity:

First paragraph (page 8, line 211): All measurements were conducted by trained researchers who followed standardized protocols to ensure consistency and accuracy in data collection. The evaluators were blinded to the participants' group assignments.

Relating to Movement Assessment Battery for Children-Second (page 9, line 225): This battery has shown strong content and construct validity for assessing motor competence in children, with test–retest reliability coefficients above 0.80 [1].

Relating to ALPHA-Fitness test battery (page 10, line 256): This battery has demonstrated high criterion validity and reliability across European school populations. Specifically, the 20-meter shuttle run, 4×10-meter shuttle run, handgrip strength test, and standing broad jump test have all been validated as reliable field-based assessments of cardiorespiratory fitness, speed/agility, and muscular strength, respectively [26,27].

Relating to NIH Toolbox software (page 12, line 289): This tool has demonstrated high internal consistency (Cronbach’s alpha > 0.80), excellent test–retest reliability, and strong convergent validity with standardized neuropsychological measures in pediatric populations [31,32].

We have revised the Conclusions section to enhance its clarity and impact, adding content on future perspectives and practical applications. The updated Conclusions section now reads as follows (page 29, line 679):

Conclusions

Our findings indicate that the association between GMC and AA in children is largely explained by improvements in CRF, S/A, and EF, with distinct mediation patterns by sex. These results emphasize the importance of implementing school-based programs that integrate motor skill development with both fitness and cognitive training. Specifically, boys may benefit more from cognitively demanding, perceptual–motor activities, while girls may respond better to interventions prioritizing physical fitness enhancement.

From a practical perspective, educators and policymakers should prioritize daily opportunities for structured physical activity that go beyond regular physical education classes, incorporating aerobic, coordinative, and cognitively engaging tasks. Such approaches could foster academic performance, reduce educational inequalities, and contribute to healthier developmental trajectories.

Future research should examine the long-term effects of integrated motor–cognitive interventions on AA, explore the neurobiological mechanisms underlying these associations, and consider additional mediators such as sleep, attention regulation, and motivation. By framing motor competence not only as a physical education outcome but as a multidisciplinary strategy, schools can promote cognitive development, academic success, and lifelong health.

Finally, numerous references in the resubmitted manuscript have been reviewed and updated to reflect more current and relevant sources. The new references have been highlighted in red, both in citation [6] and in the references list at the end of the manuscript.

Response to Reviewer 2 Comments:

Reviewer #2: Este artículo entrega respuestas a una serie de inquietudes que se presentan en el campo de la educación. La indagación tiene un correcto desarrollo que ayuda a ampliar el conocimiento respecto a la temática en cuestión. Por tanto, el presente artículo cumple correctamente las etapas en cuanto a forma y fondo.

Authors’ response: We sincerely thank you for your review and the feedback.

Response to Reviewer 3 Comments:

Reviewer #3: REVIEW REPORT (PONE-D-24-52528)

Relation between motor competence and academic achievement: the mediating role of fitness and cognition in boys and girls.

ABSTRACT

- Please adjust the acronym “AA” that was used without definition and review the presentation of the percentages and effect coefficients for adequate contextualization, focusing more on the main findings in interpretative language (for example, “cognitive mediation was more important in boys, while fitness was more expressive in girls”).

Authors' response: Thank you very much for your valuable comment. We have replaced the acronym “AA” with its full term throughout the abstract (page 2, line 67 and page 3, line 71). Additionally, we have revised the presentation of percentages and effect sizes to ensure clearer contextualization, placing greater emphasis on the main findings using more interpretative language. The revised abstract now reads as follows (page 2, line 49):

Abstract

Introduction. Gross motor competence is positively associated with academic achievement in schoolchildren, potentially mediated by fitness and cognition. However, the extent to which these mediators contribute—and whether effects differ by sex—remains unclear. This study explored the mediating roles of specific fitness components and executive function in the relationship between gross motor competence and academic achievement, considering sex differences.

Methods. This cross-sectional study included 562 Spanish schoolchildren aged 9-11 years (293 girls). Gross motor competence was evaluated using the Movement Assessment Battery for Children-Second Edition; fitness components (cardiorespiratory fitness, speed/agility, upper and lower body strength) through the ALPHA-Fitness test battery; executive function using the NIH Toolbox Battery; and academic achievement from school grades in language and mathematics. Serial multiple mediation models were applied using the PROCESS macro in SPSS, adjusted for age, BMI, and maternal education level. Analyses were conducted for the total sample and by sex.

Results. Both fitness and executive function partially mediated the relationship between gross motor competence and academic achievement. In the total sample, direct effects explained most of the association (51-73%), followed by the cognitive pathway (20-31%), fitness pathway (11-19%), and multiple pathway—gross motor competence, fitness, executive function, and academic achievement—(4-9%). Sex-specific analyses showed that cognitive mediation was predominant in boys, accounting for over half of the total effect (56–69%), with no direct effect observed. In contrast, fitness mediation was more relevant in girls, especially through cardiorespiratory fitness and speed/agility, each contributing up to 20% of the effect. The multiple pathway was also significant in girls.

Conclusions. Enhancing motor competence may improve academic outcomes, partly through gains in fitness and executive function. These findings support implementing integrated school programs, tailored to sex-specific needs—emphasizing cognitively engaging activities for boys and fitness-focused strategies for girls. The cross-sectional design implies association, not causality.

INTRODUCTION

- Overall, it is ok. But I believe that the hypothesis needs to be more evident, such as “it is expected that executive functions and fitness components partially mediate the relationship between motor competence and academic performance, with variations according to sex”.

Authors' response: Thank you for this valuable suggestion. We have incorporated a sentence stating the hypothesis at the end of the introduction, thereby strengthening the logical progression of our study and providing a natural transition to the Methods section. The final paragraph of the introduction now reads as follows (page 7, line 163):

It is therefore necessary to conduct further in-depth research into the relationships between MC, fitness components, cognitive abilities, and AA, with a particular focus on potential sex differences. The objective of this study was to examine whether fitness and EF act as mediators in the relationship between GMC and AA, and to id

---

## [Decision Letter · Decision Letter 1]

22 Oct 2025

Relation between motor competence and academic achievement: the mediating role of fitness and cognition in boys and girls

PONE-D-24-52528R1

Dear Dr. Ruiz-Hermosa,

We’re pleased to inform you that your manuscript has been judged scientifically suitable for publication and will be formally accepted for publication once it meets all outstanding technical requirements.

Kind regards,

Helmi Chaabène

Academic Editor

PLOS ONE

Additional Editor Comments (optional):

Thank you for addressing all the reviewers' comments and for revising your manuscript.

Reviewers' comments:

Reviewer's Responses to Questions

**Comments to the Author**

Reviewer #1: All comments have been addressed

Reviewer #4: All comments have been addressed

2. Is the manuscript technically sound, and do the data support the conclusions?

Reviewer #1: Yes

Reviewer #4: Yes

3. Has the statistical analysis been performed appropriately and rigorously?

Reviewer #1: Yes

Reviewer #4: Yes

4. Have the authors made all data underlying the findings in their manuscript fully available?

Reviewer #1: Yes

Reviewer #4: Yes

5. Is the manuscript presented in an intelligible fashion and written in standard English?

Reviewer #1: Yes

Reviewer #4: Yes

Reviewer #1: I would like to thank authors for this edited version of the manuscript. I believe this current version is acceptable for publication. Thank you.

Reviewer #4: (No Response)

**Do you want your identity to be public for this peer review?** For information about this choice, including consent withdrawal, please see our Privacy Policy

Reviewer #1: **Yes: ** Ferman Konukman

Reviewer #4: No

---

## [Editor Report · Acceptance letter]

PONE-D-24-52528R1

PLOS ONE

Dear Dr. Ruiz-Hermosa,

I'm pleased to inform you that your manuscript has been deemed suitable for publication in PLOS ONE. Congratulations! Your manuscript is now being handed over to our production team.

Kind regards,

on behalf of

Dr. Helmi Chaabène

Academic Editor

PLOS ONE